# Lung Organotypic Slices Enable Rapid Quantification of Acute Radiotherapy Induced Toxicity

**DOI:** 10.3390/cells12202435

**Published:** 2023-10-11

**Authors:** Maxime Dubail, Sophie Heinrich, Lucie Portier, Jessica Bastian, Lucia Giuliano, Lilia Aggar, Nathalie Berthault, José-Arturo Londoño-Vallejo, Marta Vilalta, Gael Boivin, Ricky A. Sharma, Marie Dutreix, Charles Fouillade

**Affiliations:** 1Institut Curie, Inserm U1021-CNRS UMR 3347, Paris Saclay University, Centre Universitaire, 91405 Orsay Cedex, France; 2Institut Curie, PSL Research University, 75006 Paris, France; 3SBAI Department, Sapienza University of Rome, 00161 Rome, Italy; 4Global Translational Science, Varian, a Siemens Healthineers Company, Palo Alto, CA 94304, USA; 5UCL Cancer Institute, University College London, London WC1E 6DD, UK

**Keywords:** radiation toxicity, FLASH radiotherapy, organotypic lung slices, ex vivo model, combined treatment

## Abstract

To rapidly assess healthy tissue toxicities induced by new anti-cancer therapies (i.e., radiation alone or in combination with drugs), there is a critical need for relevant and easy-to-use models. Consistent with the ethical desire to reduce the use of animals in medical research, we propose to monitor lung toxicity using an ex vivo model. Briefly, freshly prepared organotypic lung slices from mice were irradiated, with or without being previously exposed to chemotherapy, and treatment toxicity was evaluated by analysis of cell division and viability of the slices. When exposed to different doses of radiation, this ex vivo model showed a dose-dependent decrease in cell division and viability. Interestingly, monitoring cell division was sensitive enough to detect a sparing effect induced by FLASH radiotherapy as well as the effect of combined treatment. Altogether, the organotypic lung slices can be used as a screening platform to rapidly determine in a quantitative manner the level of lung toxicity induced by different treatments alone or in combination with chemotherapy while drastically reducing the number of animals. Translated to human lung samples, this ex vivo assay could serve as an innovative method to investigate patients’ sensitivity to radiation and drugs.

## 1. Introduction

In combination with chemotherapy and immunotherapy, radiotherapy remains one of the main therapeutic options in the fight against cancer [1]. To maximize the therapeutic index, radiation oncologists aim to deliver high doses to the tumor while preserving the surrounding organs at risk [2]. Several factors, such as the radiation protocol used, associated chemotherapies, or patient comorbidities, contribute to the development of radio-induced toxicities. To circumvent such toxicities, FLASH radiotherapy, an innovative radiation modality relying on ultra-high dose rates, has been recently described [3]. FLASH radiotherapy is an innovative method that delivers radiation at an ultra-high dose rate, inducing a sparing of healthy tissue from radio-induced toxicities while preserving the same anti-tumoral efficacy [3]. The so-called FLASH effect has been demonstrated in various organs, including the lung [3,4], the brain [5,6], the skin [7,8,9], and the intestine [10,11]. However, evaluation of the impact of different protocols of radiotherapy on the FLASH effect, as well as a combination with classical chemotherapies, is still needed to facilitate clinical translation of this technology. For this purpose, assays to rapidly quantify radiation toxicity of the different treatment combinations are promptly needed.

In the case of thoracic radiotherapy, radio-induced damages are characterized by pneumonitis that may evolve into lung fibrosis in the most severe cases [2]. Preclinical mouse models have been classically used to determine radiation-induced lung injury [12]. However, the number of animals required, as well as delays in fibrosis development, preclude the use of mouse models for screening purposes. Alternatively, in vitro lung models struggle to recreate the complex architecture and cellularity of lung tissue, thus lacking the cellular heterogeneity and interactions classically observed in the lung [13]. In recent years, organotypic slices have become a popular model for studying biological processes, including inflammatory responses, pathology modeling, infection, and new drug testing [14,15]. These thin tissue slices of 100 to 500 µm thickness can be obtained from murine models [16] as well as from patient samples [17,18]. In this model, cellular architecture is preserved, as shown by the characterization of epithelial cells [19,20], mesenchymal cells [21,22] (i.e., fibroblasts), endothelial cells [23], and immune populations [16,24]. Circulating immune cells are absent in organotypic lung slices, but co-culture systems have been developed to overcome this issue [25].

In the study of radiation effects, established and validated ex vivo models have been previously used in the brain [26], but, to our knowledge, it has never been applied to the lung. To evaluate the impact of different radiotherapy protocols and their association with drugs, we used lung organotypic slices to rapidly quantify radiation toxicity. First, we characterized the cell viability and cell division inside lung organotypic slices in culture. Then, a similar analysis performed 24 h after exposure to increased doses of radiation showed a dose-dependent decrease in cell viability and in the proportion of replicating cells, indicating that organotypic lung slices are a suitable model to monitor radiation toxicity ex vivo. When combined with standard chemotherapies, we showed that quantifying the proportion of replicating cells inside lung organotypic slices can be used to evaluate the toxicities of combined treatments. Interestingly, the analysis of cell replication was sensitive enough to detect a sparing effect induced by FLASH radiotherapy.

Altogether, this study demonstrates, for the first time, the usefulness of a lung ex vivo model to rapidly evaluate the toxicity of different radiation treatments (e.g., FLASH radiotherapy) while drastically reducing the number of animals required.

## 2. Materials and Methods

### 2.1. Mice and Ethics Statement

Studies were performed in accordance with the European Community recommendations (2010/63/EU) for the care and use of laboratory animals. The experimental procedures were specifically approved by the Ethics Committee of Institut Curie CEEA-IC #118 (authorization number APAFiS#32674-2021080916494690 given by the National Authority) in compliance with international guidelines. Females C57BL/6J mice purchased from Charles River Laboratories (Lyon, France) at 6 weeks of age were housed in the Institut Curie animal facilities.

### 2.2. Mouse Organotypic Lung Slices Obtention and Culture

Adult female C57BL/6J mice aged 6 to 10 weeks were anesthetized by intraperitoneal injection of ketamine/xylazine. Blood was then flushed through intracardiac injection of phosphate-buffered saline (PBS) and the trachea was exposed to inject 2 mL of 2.5% low-melting agarose (A9414-50G, Sigma-Aldrich, Saint-Louis, MI, USA) diluted in organotypic lung slices medium DMEM F12 (31331-028, Thermo Fisher Scientific, Waltham, MA, USA) supplemented with 1% SVF (CVFSVF00-0U, Thermo Fisher Scientific, Waltham, MA, USA), 1% penicillin/streptomycin (CABPES01-0U, Thermo Fisher Scientific, Waltham, MA, USA), 1% non-94 essential amino acids (11140035, Thermo Fisher Scientific, Waltham, MA, USA) and 1% L-glutamine (25030-024, Thermo Fisher Scientific, Waltham, MA, USA). Once the agarose was solidified, lungs were removed from the chest cavity, and 8 mm punches were made from each lobe individually. Tissue punches were embedded in 5% agarose, and 300 µm slices were made with a Vibratome (Leica VT1000S, Nanterre, France) as previously described [14]. The whole procedure lasted less than 2 h. Around 50 slices were obtained from a lung and placed into a 24-well plate containing each 500 µL of organotypic lung slice medium and cultured at 37 °C in 5% CO_2_ for up to 72 h.

### 2.3. Lung Slices Irradiation

Lung slices were obtained from the same biological sample and randomly assigned to different groups for the different doses of radiation in each experiment. We used the electron linear accelerator (linac) ElectronFLASH (SIT S.p.A., R&D Dept., Roma, Italy) available at Institut Curie and previously described [27]. Lung slices were irradiated in culture plates with a vertical 7-MeV beam at a source distance of 1.1 m, allowing a dose homogeneity throughout the wells’ positions better than 95%. The dosimetry was controlled by EBT-XD Gafchromic (Ashland, Bridgewater, NJ, USA) film measurements: films were cut into adapted pieces and placed at the position of the target (i.e., inside the culture wells with medium). For all conventional irradiations, we used these measurements to calibrate the monitoring ion chamber of the linac (0.007 Gy/MU at the target, which corresponds to ≈0.5 Gy/s). The linac stopped automatically when the number of Monitor Units reached the prescribed value for any dose. In the case of FLASH irradiations, we set the dose per electron-pulse at 3 Gy/pulse (±0.2 Gy) by adjusting the pulse duration, and we delivered 1, 2, and 3 pulses to achieve the target doses of 3, 6, and 9 Gy. All the beam parameters are summarized in Appendix A.

### 2.4. Whole Thorax Irradiation

Mice were exposed at the age of 10–12 weeks to a 9 or 13 Gy whole thorax irradiation with a horizontal 5-MeV beam at a Source Distance of 0.65 m, with a setup equivalent to the one previously described in [3]. Anesthesia was carried out with a nose cone using 2.5% isoflurane in the air without adjunction of oxygen. The dosimetry was controlled on an individual basis with Gafchromic films positioned on the mouse thorax surface at the center of the irradiation field. The dose rate was 0.1 Gy/s in the conventional modality and 3 Gy/pulse in the FLASH modality (Appendix A).

### 2.5. Drug Treatment

Freshly prepared organotypic lung slices were incubated at 37 °C in culture plates with concentrations of Docetaxel and Carboplatin ranging from 200 µM to 10 µM for each drug diluted in 500 µL of organotypic lung slices medium. After one hour of incubation, slices were washed twice with a medium to remove the remaining drugs. Then, slices were irradiated, and cell division was measured as described below.

### 2.6. Cell Viability

To monitor the cell death induced after irradiation in the lung slices, we stained the organotypic lung slices with Hoechst and Ethidium-1 homodimer 24 h after exposure to doses ranging from 3 to 9 Gy. Organotypic lung slices were incubated in 500 µL of culture medium containing 2 µM Ethidium-1 homodimer. Then, organotypic lung slices were washed in PBS and incubated with the Hoechst nuclear dye for 2 h. For imaging, organotypic lung slices were transferred into a glass support adapted for microscopy (µ-Slide 4 Well Glass Bottom, Ibidi, Gräfelfing, Germany) and imaged on an inverted Nikon Spinning disk TIRF-FRAP using a 20× objective. Per slice, 3 to 5 fields of view were acquired, each containing 20 stacks spaced by 3 µm. The proportion of EdU+ cells was quantified with a semi-automatic method combining 3D reconstruction and segmentation of the nuclei using IMARIS software version 9.3.1 (Bitplane, Belfast, UK).

### 2.7. Cell Replication

To estimate the proportion of cells that replicate after irradiation in the lung slices, we used a Click-IT chemistry protocol to monitor cell proliferation using 5-ethynyl-2′-deoxyuridine (EdU) incorporation (BCK-EdUPro-FC647). Irradiated slices were incubated in 500 µL of culture medium containing 10 µM EdU. After the desired incubation time (i.e., 24, 48, or 72 h), they were treated according to the manufacturer’s instructions. Then, organotypic lung slices were washed in PBS and incubated with a nuclear dye (e.g., DAPI). For imaging, organotypic lung slices were transferred into a glass support adapted for microscopy (µ-Slide 4 Well Glass Bottom, Ibidi) and imaged on an inverted Nikon Spinning disk TIRF-FRAP using a 10× objective. Per slice, 3 to 5 fields of view were acquired, each containing 50 stacks spaced by 3 µm. The proportion of EdU+ cells was quantified with a semi-automatic method combining 3D reconstruction and segmentation of the nuclei using IMARIS software Bitplane. 

### 2.8. Statistical Analysis

Statistical analyses were performed using the ggpubr package (https://rpkgs.datanovia.com/ggpubr/ (accessed on 4 September 2023)) in R. For our EdU+ nuclei count data, we assumed samples are not independent (from the same biological sample), and our data do not follow a normal distribution. Comparisons of the means of these counts for the FLASH and CONV.

## 3. Results

### 3.1. Characterization of Lung Organotypic Slices

To counteract the limitations of current in vitro assays for radiation studies in the lung, we aimed to implement an ex vivo lung model to enable rapid and quantitative analysis of radiation toxicity. In order to preserve the complexity of the lung architecture and the distinct cell types, we adapted the lung ex vivo model, previously used for chemical toxicity study, to investigate radiation toxicities in healthy lungs. Briefly, mouse lungs were inflated with agarose and sliced at 300 µm. Once obtained, lung slices, maintained in a medium, were irradiated and analyzed in the days following their preparation (Figure 1A). To evaluate the preservation of lung architecture and, in particular, alveoli structure in the slices, we crossed the mouse transgenic line Sftpc-CreERT2 [28] with the reporter line carrying the R26-mTmG allele [29] to label the AT2 cells with GFP upon tamoxifen treatment. This analysis showed that the lung structure is maintained in the slices and that AT2 cells are nicely preserved in the alveoli.

### 3.2. Analysis of Cell Viability and Cell Division in Organotypic Lung Slices

To characterize the lung ex vivo model, we aimed to evaluate the cell viability and cell division of organotypic slices for the first 3-days after obtention. For these purposes, we quantified, on the one hand, the proportion of dead cells stained by Ethidium (Figure 2A) and, on the other hand, the proportion of replicating cells that incorporate EdU for a period of 24, 48, or 72 h (Figure 2B). The analysis of cell viability showed that the proportion of Ethidium-positive cells increased over time from 14.9% after 24 h, 27.5% after 48 h, and 45.2% after 72 h (Figure 2C). On a similar trend, the proportion of replicating cells per field of view (FOV) increased steadily from 36 EdU+ cells after 24 h to 480 cells after 48 h and 818 cells after 72 h (Figure 2D).

Interestingly, the analysis of EdU incorporation in the lung by flow cytometry revealed that between 0.5 and 1% of cells are replicating in vivo for 24 h, suggesting that ex vivo analysis of EdU incorporation at 24 h reflects physiological conditions occurring in vivo (Appendix A). For further analysis, we selected the 24 h time point to evaluate radiation toxicity in lung organotypic slices.

### 3.3. Radiation Induces a Dose-Dependent Decrease in Cell Viability and Cell Proliferation Ex Vivo

To characterize the effect of radiation in organotypic lung slices, we first monitored cell viability after exposure to different doses of radiation using Ethidium-1 staining (Figure 3A,C). Quantification of Ethidium-1 positive cells in organotypic lung slices 24 h after radiation exposure showed a dose-dependent increase in the proportion of Ethidium-1 positive cells ranging from 37.7% after 3 Gy to 44.8% after 6 Gy and 50.2% after 9 Gy. This result confirms that radiation induces ex vivo a dose-dependent decrease in viability.

Because it is well known that radiation triggers cell cycle arrest to enable cell repair [2], we hypothesized that radiation may impact the capacity of lung cells to replicate after irradiation. To test this hypothesis ex vivo, organotypic lung slices were exposed to conventional radiation at doses ranging from 3 Gy to 9 Gy and immediately incubated with EdU for 24 h. As expected, the proportion of EdU+ cells progressively decreased as the dose increased (Figure 3B,D). Quantification of the proportion of EdU+ cells after exposure to doses of 3 Gy, 6 Gy, and 9 Gy showed a decrease of 53%, 68%, and 80%, compared to the proportion of EdU+ cells observed in non-irradiated slices. To evaluate the robustness of the method, we estimated the variability in the proportion of EdU+ cells quantified after a dose of 9 Gy (i) between technical replicates by comparing four different organotypic slices obtained from the same mouse (Appendix A) and (ii) between biological replicates by analyzing organotypic slices obtained from three different mice (Appendix A). No significant difference was observed between the slices from the same mouse or between mice.

Altogether, these results indicate that radiation induces ex vivo a dose-dependent decrease in cell viability as well as cell replication 24 h after irradiation. Considering the robustness and reproducibility of the analysis of EdU incorporation in organotypic lung slices, these analyses appear to be suitable for comparison between different radiation modalities.

### 3.4. Cell Replication Analysis Ex Vivo Allows to Discriminate between FLASH and CONV Irradiation

To determine if ex vivo analysis of cell replication is sufficient to discriminate between conventional (CONV) and FLASH modalities, we quantified the proportion of EdU+ cells in lung organotypic slices after exposure to different doses delivered either in CONV or FLASH conditions using the ElectronFLASH linear accelerator developed by SIT company (SIT S.p.A., R&D Dept., Roma, Italy). As a new FLASH device, we first confirmed that whole thorax FLASH irradiation using the ElectronFLASH linac triggers a sparing from radiation-induced lung fibrosis (Figure 4A,B). Then, we exposed the organotypic lung slices to doses of 3 Gy, 6 Gy, and 9 Gy delivered either in CONV (0.5 Gy/s) or FLASH (>100 Gy/s) modes and quantified the proportion of EdU+ cells 24 h after irradiation (Figure 4C,D and Appendix A). Interestingly, the proportion of replicating cells was higher after FLASH than CONV irradiation for the doses of 6 Gy and 9 Gy. These encouraging results indicate that analysis of EdU incorporation is suitable for detecting a FLASH-sparing effect on healthy tissue ex vivo.

To further validate the relevance of the cell replication potential to monitor the FLASH effect, we monitored in vivo the proportion of replicating cells 24 h after whole thorax exposure to a dose of 9 Gy delivered either in CONV or FLASH conditions (Figure 5). Despite some variability in the proportion of cycling cells between mice, FLASH thoracic irradiation induced a higher proportion of EdU+ cells than after CONV irradiation (Figure 5B,C), confirming the relevance of the measurement of cell replication to evaluate the FLASH effect ex vivo. Altogether, these results show that analyzing the proportion of cycling cells 24 h after irradiation is a rapid and efficient method to evaluate the FLASH effect on healthy lungs.

### 3.5. Organotypic Lung Slices Assay Can Evaluate Combined Treatment Toxicity

Most patients are treated by chemotherapies before radiation, and with the development of new drugs, there is a need to evaluate the toxicity of combined treatments on healthy organs. As a proof of concept for the treatment of lung cancer, we evaluated the combination of cytotoxic chemotherapies, such as Carboplatin and Docetaxel, with conventional radiotherapy. To determine the optimal concentration for each drug, we incubated the lung slices for 1 h with different doses of Carboplatin or Docetaxel, washed the drug, and incubated the slices with EdU. Quantification of the cycling cells 24 h after drug treatment confirmed, for both chemotherapeutic agents, a dose-dependent decrease in the proportion of EdU+ cells (Figure 6A,B).

Then, to evaluate radiation toxicity following drug treatment, we selected doses of Carboplatin and Docetaxel, inducing a slight decrease in the proportion of cycling cells, respectively 100 µM and 50 µM, and chose a conventional radiotherapy dose of 2 Gy to estimate the effect of a classical fraction of radiotherapy. Analysis of the proportion of replicating cells indicates that a single treatment of either 2 Gy or Carboplatin or Docetaxel alone induces a respective decrease of 31.2%, 29.9%, or 57.4% compared to non-treated samples. Interestingly, a combination of chemotherapies with radiation led to a more pronounced decrease in replicating cells, with 51.6% for Carboplatin-treated slices and 70.6% for Docetaxel-treated slices (Figure 6C,D). In conclusion, quantification of cell replication ex vivo using EdU incorporation allows us to rapidly evaluate the toxicity induced by a combination of cytotoxic drug treatments with radiation, potentially paving the way to preclinical investigations of a wide range of chemotherapies in association with radiotherapy.

## 4. Discussion

The aim of this study was to investigate ex vivo lung organotypic slices as a model to robustly and rapidly evaluate radiation toxicity in the lung. As a first step, we characterized the model showing that (i) the lung and alveolar structure is preserved in the slices, (ii) the optimal time point for analysis is 24 h after obtention when slice viability is the highest, and the percentage of replicating at 24 h post-irradiation is consistent with the proliferation rate observed in vivo. After exposure of the slices to increased doses of radiation, we observed a dose-dependent decrease in cell viability as well as in the proportion of cells able to incorporate EdU, in accordance with what is known of the effects of radiation on tissue. Importantly, quantification of the proportion of dividing cells into the slices after exposure to doses of 6 Gy or 9 Gy delivered either in conventional or FLASH modalities was sufficiently sensitive and robust to detect a FLASH-sparing effect (i.e., a higher proportion of replicating cells after FLASH compared to conventional irradiation). To potentiate the translational relevance of the findings, we demonstrated that the measurement of replicating cells in this ex vivo model can be used to assess the impact on healthy lungs of combined treatments such as the combination of chemotherapy with radiation. Preclinical studies of radiotherapy classically require large cohorts of animals, and the lung ex vivo model to analyze treatment toxicity is an interesting alternative to reduce the number of animals used in medical research.

Regarding the growing interest in FLASH radiotherapy in the community, the results obtained with the ex vivo lung model are in accordance with the known effects of FLASH irradiation on healthy tissue. Indeed, it has been previously shown in the brain and in the gut that FLASH radiotherapy spares the number of neuronal precursors in the dentate gyrus as well as the proportion of dividing stem cells from the intestinal crypts [5,10]. Previous studies in the lung reported a higher proportion of EdU+ cells 1-week after conventional compared to FLASH [4]. This discrepancy may be explained by the difference in EdU exposure (i.e., 24 h vs. daily injection during one week) and time of analysis after irradiation (i.e., 24 h vs. 1 week). One explanation could be that the present study investigates the cell cycle arrest induced 24 h after CONV and FLASH irradiation, while previous analysis integrates all the replicating cells in the week following irradiation, probably including the compensatory proliferation required to replace the damaged lung cells. More importantly, because of the rapid development of FLASH clinical trials, like the FAST-01 nonrandomized trial for the treatment of symptomatic bone metastases at the Children’s Hospital in Cincinnati (USA) [31], clinical investigations are starting in different centers, and there are important needs to validate the FLASH capabilities of new irradiation devices as well as to determine the optimal biological parameters to observe the FLASH sparing effect. The robust analysis of dividing cells in lung organotypic slices can serve to rapidly assess if a new setup or new beam parameters are capable of inducing a FLASH-sparing effect in healthy tissue. In addition, this ex vivo model can reduce the cost of challenging FLASH studies while saving time in animal studies (i.e., results are obtained within days compared to several months when analyzing late toxicity, such as pulmonary fibrosis).

As a proof of concept, we have also shown that analysis of cell replication ex vivo can estimate lung toxicity induced by the association of radiation with drugs such as Carboplatin and Docetaxel, two drugs routinely used for the systemic treatment of lung cancer [32]. While lung organotypic slices have been previously used for the evaluation of respiratory toxicity induced by industrial compounds [33], it is the first time that a lung ex vivo model and, in particular, the count of EdU+ cells, is implemented as a preclinical tool for the assessment of toxicities induced by chemotherapies and radiation. Our encouraging results reported in this study pave the way to translational studies aiming at evaluating innovative radiation modalities alone or in combination with anti-cancer drugs to prevent lung toxicities (i.e., pneumonitis, interstitial fibrosis) affecting patients’ quality of life [34]. 

This work provides an analysis of cell viability and replication after radiation, but this lung ex vivo model requires further characterization to integrate a more detailed analysis of the multiple processes induced by radiotherapy. Indeed, radiation modality such as stereotactic radiotherapy using high doses per fraction for the treatment of early-stage lung tumors and metastasis triggers activation of a diverse set of compartments such as endothelial, mesenchymal, and immune cells. A deeper analysis of these specific populations will provide a more precise analysis of the acute effects of radiation in the lungs. More broadly, to provide a more comprehensive understanding of radiation injury in the lung, one could benefit from modern computational methods, like those used previously [35], to model the impact of radiation ex vivo.

We have shown that lung ex vivo models are suitable to rapidly analyze radiation toxicities, but alternative methods may also be considered. Indeed, future computational models of lung radiation injury can serve to predict responses to different treatments (i.e., radiotherapy alone or in combination with drugs). In addition, non-invasive imaging techniques (e.g., SPECT, CT scans) and the use of biomarkers to detect oxidative stress and endothelial cell death in vivo can provide early detection methods to estimate lung responses to radiotherapy [36]. 

Another limitation of this study is the use of organotypic slices prepared from mouse lungs. Alternatively, slices can be prepared from human lung resected from lobectomy [15,37], and future work will concentrate on the development of lung organotypic slices from patients to investigate treatment toxicities in humans. This will pave the way to more clinically relevant investigations on radiation toxicities, taking into account patients’ intrinsic radiosensitivity. 

## Figures and Tables

**Figure 1 cells-12-02435-f001:**
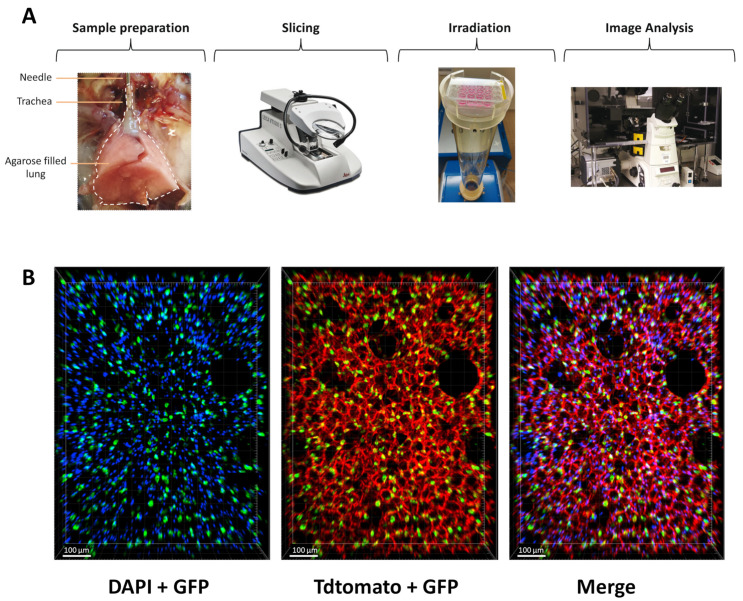
Structure and morphology of the lung are preserved ex vivo. (**A**) Scheme of the protocol to obtain and analyze radiation toxicity using lung organotypic slices. (**B**) Lung organotypic slices prepared from Sftpc-CreERT2; R26-mTmG mice in which all cells expressed TdTomato allowed to visualize the preserved structure of the lung and AT2 cells, labeled with GFP are present in the alveoli.

**Figure 2 cells-12-02435-f002:**
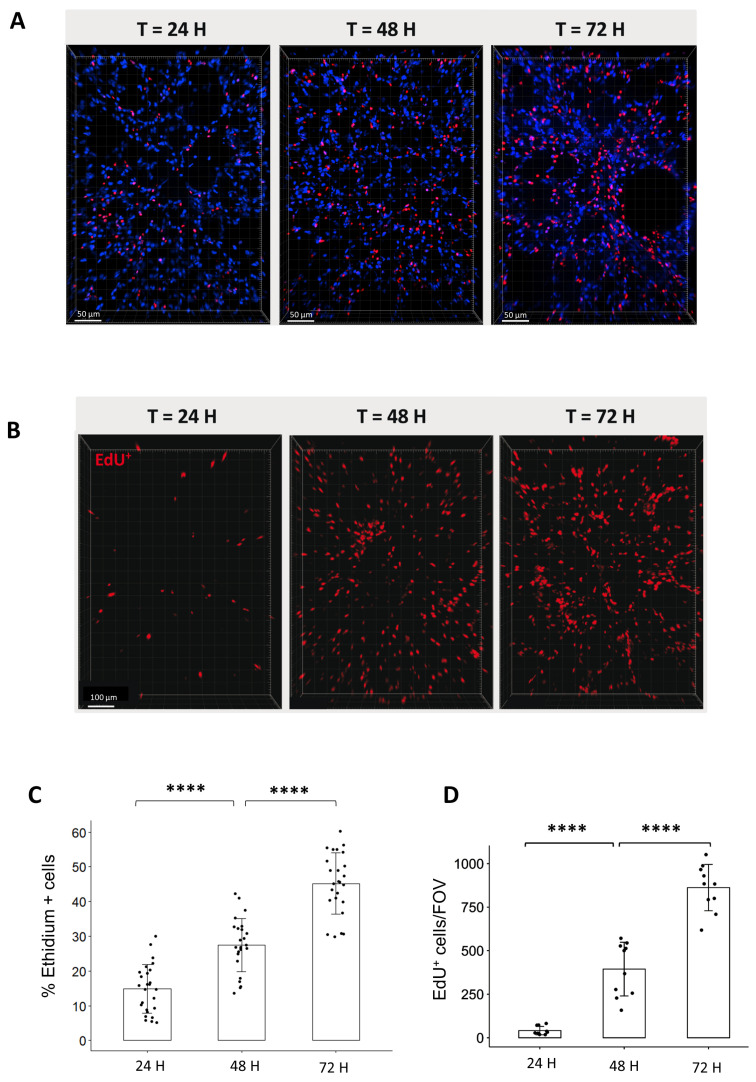
Analysis of cell viability and replication in organotypic lung slices in culture. (**A**) Representative 3D reconstruction of Ethidium incorporation in organotypic lung slices after 24, 48, and 72 h in culture. Images were acquired on a confocal microscope with a 20× objective. The scale bar represents 50 µm. (**B**) Representative 3D reconstruction showing EdU+ cells after exposure to EdU for either 24, 48, or 72 h. Images were acquired on a confocal microscope with a 10× objective. The scale bar represents 100 µm. (**C**) The quantification of the proportion of Ethidium positive cells per FOV (4 to 5 FOV for *n* = 5 slices) showed an increase in the proportion of dead cells from 24 to 72 h. Each dot represents the quantification for one field of view (**D**) Image analysis and quantification showed a steep increase in the number of EdU^+^/FOV over time (min 4 FOV for *n* = 3 slices). Each dot represents the quantification for one field of view. **** *p*-value < 0.0001.

**Figure 3 cells-12-02435-f003:**
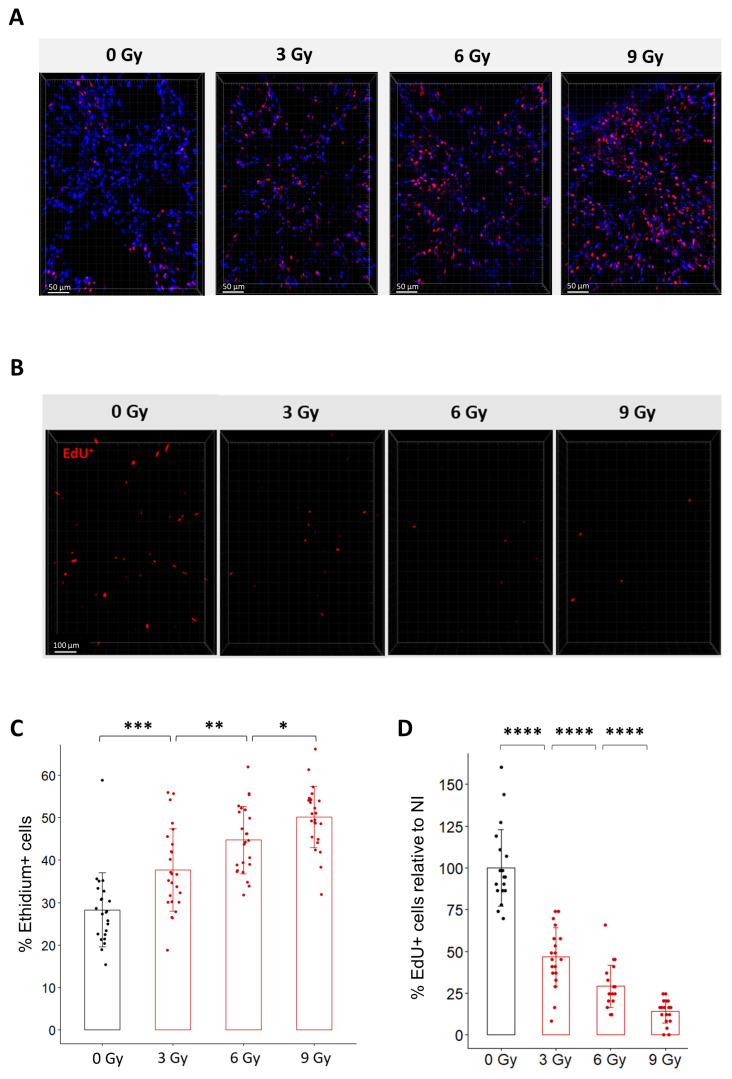
Dose-dependent effects of radiation on cell viability and cell replication in organotypic lung slices. (**A**) Representative 3D reconstruction of cell viability analysis in organotypic lung slices 24 h after exposure to different doses of radiation. Images were acquired on a confocal microscope with a 20× objective. The scale bar represents 50 µm. (**B**) Representative 3D reconstruction showing EdU+ cells 24 h after exposure to doses from 3 to 9 Gy dose of radiation. Images were acquired on a confocal microscope with a 10× objective. The scale bar represents 100 µm. (**C**) Quantification and image analysis revealed a dose-dependent increase in the proportion of Ethidium-positive cells per FOV (*n* = 4 to 5 FOV per slice for a total of 5 slices per condition) Each dot represents the quantification for one field of view. (**D**) Image analysis and quantification show a significant dose-dependent decrease in the proportion of EdU+ cells 24 h after radiation exposure (*n* = 4 to 5 FOV per slice for a total of 5 slices per condition) Each dot represents the quantification for one field of view. * *p*-value < 0.05; ** *p*-value < 0.01; *** *p*-value < 0.001; **** *p*-value < 0.0001.

**Figure 4 cells-12-02435-f004:**
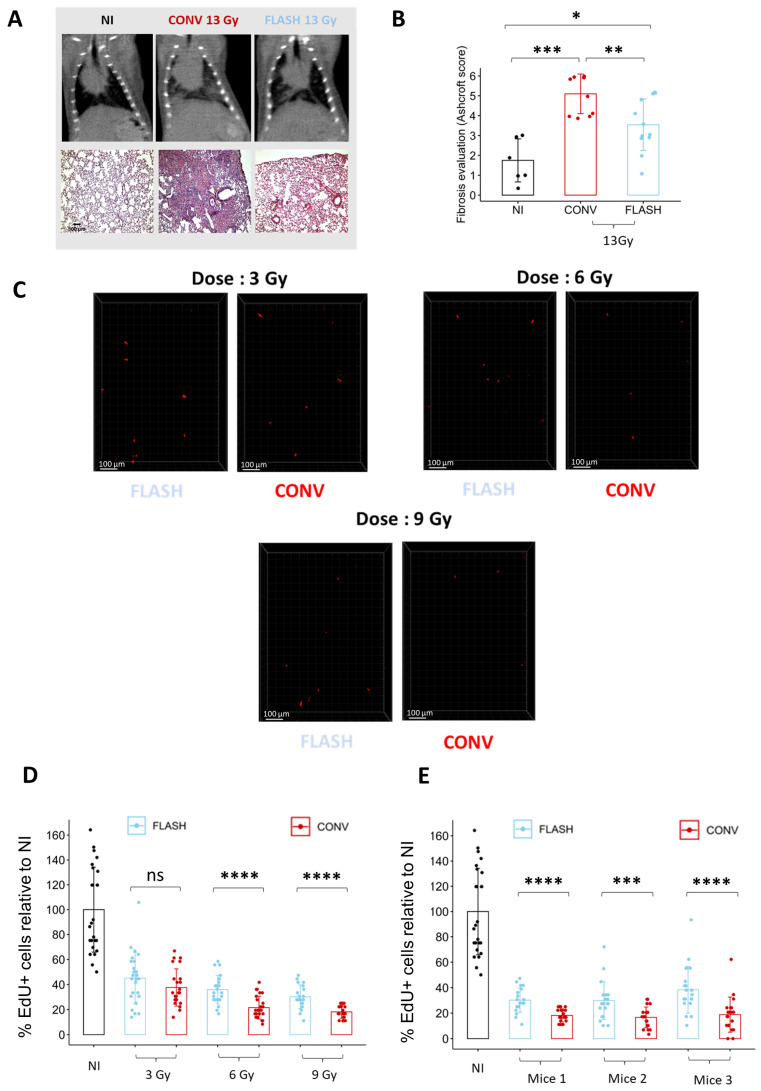
Ex vivo analysis of cell replication allows to discriminate between FLASH and CONV irradiation. (**A**) Representative CT-scan images from C57BL/6J mice 20 weeks after 13 Gy irradiation and hematoxylin/eosin/saffron staining (HES) of lung sections, resected 20 weeks after 13 Gy thoracic irradiation, showing thickening of alveolar septa, collagen deposition, and large fibrotic areas after CONV irradiation. Lung fibrosis development was reduced after FLASH. (**B**) Histopathological analysis of lung fibrosis using the Ashcroft score [30] revealed more severe fibrosis after CONV than FLASH irradiation. (**C**) Representative 3D reconstruction showing EdU+ cells 24 h after exposure to a range from 3 to 9 Gy dose of conventional or FLASH radiation. Images were acquired on a confocal microscope with a 10× objective. The scale bar represents 100 µm. (**D**) Cell division analysis using the EdU assay revealed a higher proportion of EdU+ cells after FLASH compared to CONV irradiation for doses of 6 Gy and 9 Gy (5 FOV for *n* = 5 slices) Each dot represents the quantification for one field of view. (**E**) Quantification of the proportion of EdU^+^ cells in three independent mice confirmed the robustness of cell division assay to detect a short-term FLASH effect. (5 FOV for *n* = 5 slices) Each dot represents the quantification for one field of view. ns, not significant; * *p*-value < 0.05; ** *p*-value < 0.01; *** *p*-value < 0.001; **** *p*-value < 0.0001.

**Figure 5 cells-12-02435-f005:**
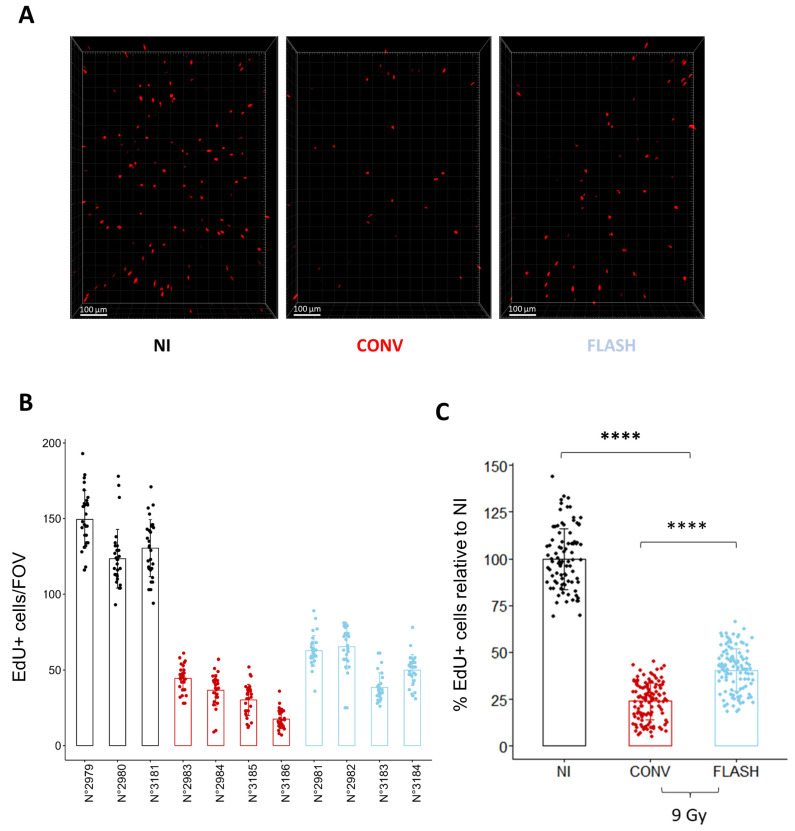
Cell replication analysis in vivo confirms the FLASH-sparing effect observed ex vivo. (**A**) Representative 3D reconstruction showing EdU+ cells 24 h after whole thorax irradiation at a dose of 9 Gy delivered either in CONV or FLASH modalities. Images were acquired on a confocal microscope with a 10× objective. The scale bar represents 100 µm. (**B**) Quantification of the number of EdU+ cells/FOV per mouse shows inter-individual variability of the proportion of replicating cells between mice. Each dot represents the quantification for one field of view (**C**) Grouped per condition and normalized to the proportion of replicating cells in the non-irradiated control, FLASH spared a significant proportion of replicating cells in vivo, confirming the results obtained ex vivo. (*n* = 3 to 4 mice per condition for *n* = 5 slices per mouse and 5 FOV per slice) Each dot represents the quantification for one field of view. **** *p*-value < 0.0001.

**Figure 6 cells-12-02435-f006:**
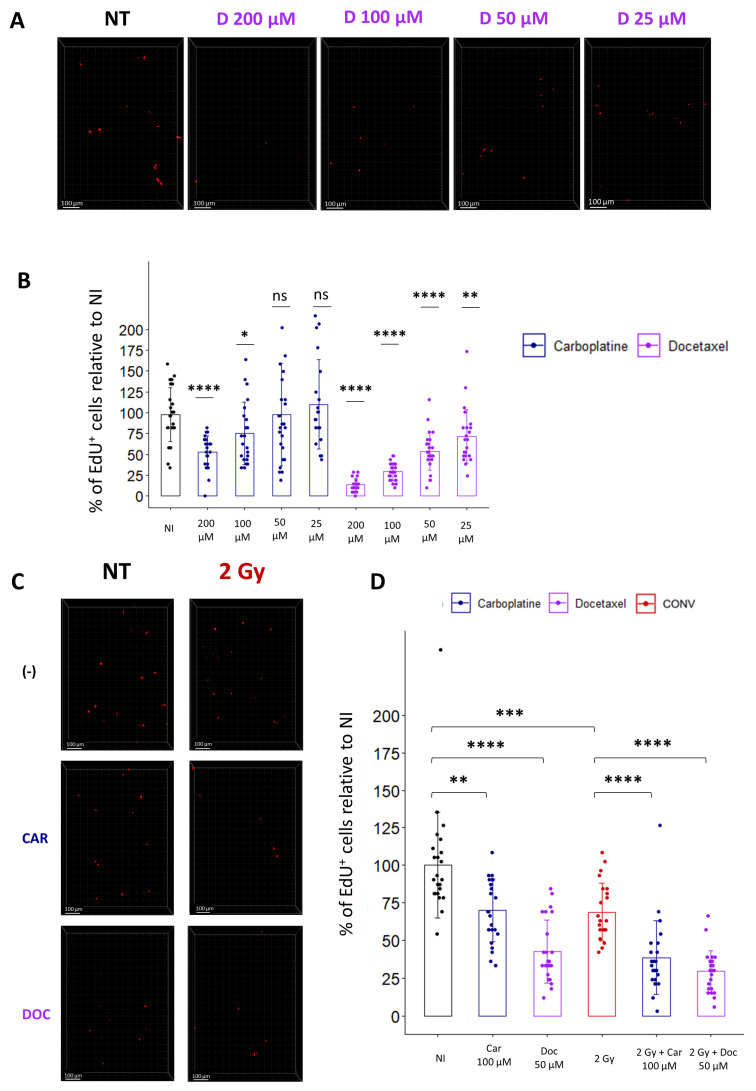
Ex vivo analysis of cell replication allows the evaluation of toxicity induced by chemotherapies alone or in combination with radiation. (**A**) Representative 3D reconstruction showing EdU+ cells 24 h after exposure to different concentrations of Docetaxel (from 25 µM to 200 µM). Images were acquired on a confocal microscope with a 10× objective. The scale bar represents 100 µm. (**B**) Image analysis and quantification show a significant decrease in the proportion of EdU+ cells 24 h after chemotherapy exposition (5 FOV for *n* = 5 slices) Each dot represents the quantification for one field of view. (**C**) Representative 3D reconstruction showing EdU+ cells 24 h after exposure to either a chemotherapeutic agent alone (i.e., Docetaxel 50 µM or Carboplatin 100 µM) or in combination with radiation (2 Gy). Images were acquired on a confocal microscope with a 10× objective. (**D**) Images quantification of the proportion of EdU+ cells reveals that chemotherapies alone induce a significant decrease in the proportion of replicating cells that is accentuated when chemotherapies are associated with radiation (5 FOV for *n* = 5 slices) Each dot represents the quantification for one field of view. ns, not significant; * *p*-value < 0.05; ** *p*-value < 0.01; *** *p*-value < 0.001; **** *p*-value < 0.0001.

## Data Availability

All data generated and analyzed during this study are included in this published article (and its Appendix A). Raw data will be shared upon request to the corresponding author.

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
