# Peer review of "Lung Organotypic Slices Enable Rapid Quantification of Acute Radiotherapy Induced Toxicity"

_cells, 2023, doi:10.3390/cells12202435_

Round 1

Reviewer 1 Report

This manuscript describes and demonstrates the usefullness of organotypic slices to study normal lung toxicity following radiation therapy/chemotherapy exposures. This is a convincing and well-written study. I have nevertheless several remarks that may be taken into account before publication.

1- In the introduction section, line 34, concerning factors influencing radiation toxicity, one may include the notion of pre-existing co-morbidities before intrinsic radiation sensitivity, the later being quite difficult to evaluate. Line 40, why omitting studies published on the FLASH effect in the lung? This should be added in the citations.

2- In figure 2 and 3, do authors have an idea of which cells are dying in their model, spontaneously and in response to radiation exposure?

3- Figure 4: are there any data on the FLASH effect on cell viability? Are the data in figure 4D CONV the same as those presented in figure 3D?

4- Figure 5: data show a FLASH effect, preserving replicative EdU+ cells. This is not in accordance with previously published data in Clinical Cancer Research (2020), demonstrating reduced percentage of EdU+ cells following FLASH irradiation versus CONV. This does not weaken the present study but should be evoked in the discussion section. Do authors have an idea of what kind of cells are cycling? 

5- The pertinence of this model is based on cell viability/cell death in response to radiation exposure/chemotherapy to evaluate lung toxicity. This are convicing results. However, and this should be evoked in the discussion section, lung response to IR, especially in response to modern radiation therapies such as severe hypofractionation for example in cases of SBRT for early stage NSCLC or lung metastasis, implies multiple processes of cell activation (endothelial, mesenchymal, immune) and probably not only cell replication capacity/death. This is a beautifull model but these limits should be discussed.

Reviewer 2 Report

The study was focused on introducing new method of quantification of radiotheraphy induced toxicity in lungs. Authors described methodology which is very innovative, and it is based on an ex vivo experiments which may be very important in the context of reducing the number of animals employed for experiments (3R rule). To evaluate the impact of different radiotherapy protocols and their association with 63 drugs, authors used lung organotypic slices from normal mice and maintained these pieces of tissue to be able to estimate cell viability and cell division after treatment.

The structure of the paper is correct, and conclusions are clear and supported by well-set experiments. However, I have two points that need to be explained:

1/ Have authors ever done experiments comparing ex vivo with in vivo? It would be very valuable to see the effect of ratiotheraphy treatment on lung tissue in vivo. I do not recommend doing it for all experimental setups included in the paper but only for so called positive and negative controls. If such experiments were done before, please include information in the discussion section.   

2/ Authors used Hoechst 33342 dye to visualize live cells. Have you considered using Tunnel assay to visualize cell apoptosis?

Reviewer 3 Report

The paper presents a study focused on the utilization of ex vivo lung organotypic slices as a model for assessing the toxicity of radiation therapy. The authors suggest this model as an alternative to traditional animal testing for evaluating lung damage resulting from radiation therapy alone or in combination with chemotherapy. 

A notable criticism of this study is its reliance on a somewhat simplified model that attempts to predict radiotherapy outcomes using simple variables (cell viability). While the authors have conducted experiments establishing a connection between radiotherapy-induced toxicity and cell viability/cell division, it's important to note that the correlation between radiation dosage and cell viability is a well-established concept. Consequently, it becomes crucial for this paper to delve into several key aspects:

Mechanistic Insights: The paper should discuss deeper into the underlying mechanisms responsible for the observed correlation. This includes exploring the molecular and cellular pathways through which radiation affects cell viability. In this context,  modern computational mechanistic models [1] and deep learning models [2] employ complex, large-scale models with millions of parameters to provide a more reasonable understanding of lung cellular injury.

Comparison with Alternative Methods: The paper should present a comprehensive discussion regarding the trade-offs between the proposed ex vivo lung organotypic slice model and alternative non-invasive methods. This should encompass computational models mentioned above, and imaging approaches (including CT scans). Some discussion of the CT methods are included in the paper,  the authors argued that fibrosis occurs at a very late stage and thus exclude CT for screening purposes. However, it should be acknowledged the presence of biomarkers [3] and early detection methods for assessing lung injury before the onset of fibrosis.

In summary, the study's robustness and relevance could be significantly improved by addressing these points. This would provide a more comprehensive understanding of the potential applications and limitations of the ex vivo lung organotypic slice model in the context of radiation therapy toxicity assessment.

Minor:

  • The text “*** μm” on some of the figure scale bars are too small to be visible. 

  • Similarly, some the p values for statistical tests are too small. For example, p values on Figure 3C are OK but on Figure 3D they are not visible

  • The study should discuss how this established correlation can be applied practically. Is there a discernible threshold value that determines a radiation dose's acceptability for radiotherapy? Discuss the balance between therapeutic benefits (tumor control) and toxicity (normal tissue damage).

  • Species Differences: If the study employs animal (mice) models to evaluate lung injury relevant to humans, it is imperative to address the known differences in radiation tolerance between mice lung cells and human lung cells. Explain any existing methods for adjusting findings from animal models to be applicable to human patients.

References

  1. https://www.frontiersin.org/articles/10.3389/fphys.2019.00191/full

  3. Audi, S. H., Jacobs, E. R., Zhang, X., Camara, A. K. S., Zhao, M., Medhora, M. M., Rizzo, B., & Clough, A. V. (2017). Protection by Inhaled Hydrogen Therapy in a Rat Model of Acute Lung Injury can be Tracked in vivo Using Molecular Imaging. Shock (Augusta, Ga.), 48(4), 467–476. https://doi.org/10.1097/SHK.0000000000000872

Round 2

Reviewer 3 Report

After carefully reviewing the revised manuscript, I believe the authors have sufficiently addressed the major issues. Therefore, I recommend that the manuscript be published at this time